# *Acidovorax citrulli* Effector AopV Suppresses Plant Immunity and Interacts with Aromatic Dehydratase ADT6 in Watermelon

**DOI:** 10.3390/ijms231911719

**Published:** 2022-10-03

**Authors:** Jie Jiang, Mei Zhao, Xiaoxiao Zhang, Linlin Yang, Nuoya Fei, Weiqin Ji, Wei Guan, Ron Walcott, Yuwen Yang, Tingchang Zhao

**Affiliations:** 1State Key Laboratory for Biology of Plant Diseases and Insect Pests, Institute of Plant Protection, Chinese Academy of Agricultural Sciences, Beijing 100193, China; 2Department of Plant Pathology, College of Plant Protection, China Agricultural University, Beijing 100193, China; 3Department of Plant Pathology, University of Georgia, Athens, GA 30602, USA

**Keywords:** *Acidovorax citrulli*, bacterial fruit blotch, effectors, AopV, ADT6

## Abstract

Bacterial fruit blotch (BFB) is a disease of cucurbit plants caused by *Acidovorax citrulli*. Although *A. citrulli* has great destructive potential, the molecular mechanisms of pathogenicity of *A. citrulli* are not clear, particularly with regard to its type III secreted effectors. In this study, we characterized the type III secreted effector protein, AopV, from *A. citrulli* strain Aac5. We show that AopV significantly inhibits reactive oxygen species and the expression of PTI marker genes, and helps the growth of *Pseudomonas syringae* D36E in *Nicotiana benthamiana*. In addition, we found that the aromatic dehydratase ADT6 from watermelon was a target of AopV. AopV interacts with ADT6 in vivo and in vitro. Subcellular localization indicated ADT6 and AopV were co-located at the cell membrane. Together, our results reveal that AopV suppresses plant immunity and targets ADT6 in the cell membrane. These findings provide an new characterization of the molecular interaction of *A. citrulli* effector protein AopV with host cells.

## 1. Introduction

Plants have an innate immune system to defend themselves against invasion and colonization by plant pathogens [1]. On the plant cell surface, receptor kinases and receptor-like proteins act as pattern recognition receptors (PRRs) to sense foreign microbial molecules, called pathogen-associated molecular patterns (PAMPs) or microbial-associated molecular patterns (MAMPs). When pathogens infect plants, they may trigger two layers of defense responses. The first layer is the PAMPs/MAMPs-triggered immunity (PTI) [2,3]. The PTI responses are mainly manifested as the expression of defense-related genes, reactive oxygen species (ROS) burst, callose deposition, etc. The second layer is the effector-triggered immunity (ETI). The ETI includes accelerated and amplified PTI responses, which are related to hypersensitive response (HR) and systemic acquired resistance (SAR) [1].

Many plant pathogenic bacteria use the type III secretion system (T3SS) to secrete effector proteins into plant cells; these effectors help bacterial cells proliferate in the host [4]. Type III secreted effectors (T3Es) suppress plant immune responses using various strategies, which include interference with signal transduction of immune pathways, blocking RNA pathways and vesicle transport, changing organelle functions, and inhibition of plant PTI responses [5,6]. For example, *Pseudomonas syringae* T3E HopAI1 inhibits *Arabidopsis thaliana* mitogen-activated protein kinases (MAPKs) activated by exposure to PAMPs [7]. The *Xanthomonas oryzae* T3E XopL activates ROS accumulation in the non-host plant *Nicotiana benthamiana* by mediating the degradation of ferredoxin [8]. Effectors AvrPto and AvrPtoB, from *P. syringae*, target BIK1 in *A. thaliana* to prevent its signal transduction [9]. However, T3Es also trigger the ETI responses during incompatible plant-pathogen interactions. For example, XopQ from *Xanthomonas oryzae* pv. *oryzae* triggers ETI in the non-host plant *N. benthamiana* [10] and interacts with the resistance protein Roq1 in *N. benthamiana*, resulting in an HR [11]. The immune responses during PTI and ETI are interconnected, but different [12]. ETI and PTI are not two distinct lines of defense [13]. The occurrence of the ETI responses in *A. thaliana* also depends on various elements in the PTI pathways [14].

After the pathogen attack, salicylic acid (SA), jasmonic acid (JA), ethylene, and other plant hormones play an important role in the activation of plant defense responses [15,16]. SA mainly plays a role in the expression of defense-related genes and SAR [17]. For example, transgenic tobacco and *A. thaliana* that express bacterial salicylate hydroxylase cannot accumulate SA. This not only prevents plants from inducing SAR, but also increases their susceptibility to pathogens, such as viruses, fungi, and bacteria [18,19]. A recent study showed that *A. citrulli* T3E AopP targets the watermelon transcription factor ClWRKY6 and suppresses SA signaling [20]. With regards to JA, it mainly protects the host from necrotrophic pathogens by inducing systemic resistance [21]. Therefore, JA synthesis-related genes are also targets of pathogenic microorganisms. For example, the *P. syringae* T3E HopZ1a interacts with the transcription repressor ZIM domain protein to activate JA signaling [22]. *Pseudomonas syringae* T3E HopBB1 interacts with *A. thaliana* transcription factor TCP14 to de-repress JA response genes [23]. However, no studies have reported *A. citrulli* T3Es affecting JA. 

Bacterial fruit blotch (BFB) of cucurbits has occurred worldwide, possibly via seed transmission [24]. *Acidovorax citrulli*, the causal agent, can infect a variety of Cucurbitaceae crops, especially watermelon and cantaloupe, and cause severe economic losses. Despite this, the molecular interactions between *A. citrulli* and host plants are not well-defined, and effective BFB management methods, including commercial sources of resistance, are lacking. At present, the research on *A. citrulli* mainly focuses on type II, type III, and type VI secretion systems, among which the research on the T3SS has made progress [25,26,27,28]. The T3SS is a key pathogenicity factor of *A. citrulli* [26]. The T3SS apparatus gene *hrpE* encodes for the Hrp pilus and serves as a conduit to secrete effector proteins into host cells [29]. *HrpG* and *hrpX* are key regulatory genes of T3SS in *A. citrulli* [30]. In addition, N. benthamiana can be used to study *A. citrulli* T3Es [31]. Comparing the differences between the *A. citrulli* wild-type strain M6 and the hrpX mutant strain, some T3Es were successfully identified [32]. However, research on *A. citrulli* T3Es is still relatively limited. Detailed *A. citrulli* T3E characterization showed that AopP suppressed SA signaling and AopN caused programmed cell death in plants. Both AopP and AopN interfered with the plant immune responses [20,33]. However, other effector proteins interfering with plant immunity in *A. citrulli* have not been reported yet. Exactly, how *A. citrulli* T3Es interfere with plant immunity is still largely unknown.

In this study, we identified AopV as a T3E of *A. citrulli*, and found it was localized in the plant host cell nucleus and cell membrane. Importantly, the effector significantly inhibited the PTI responses in *N. benthamiana*. In addition, AopV activated the JA pathways. This study provides new insights into the pathogenicity mechanism of *A. citrulli*.

## 2. Results

### 2.1. Sequence Analysis of AopV in A. Citrulli Strain Aac5

The DNA sequence of *aopV* from Aac5 is 100% identical to the homologous genes from *A. citrulli* group I strain M6 (genome accession number CP029373.1) and AAC00-1 (genome accession number CP000512.1). The putative effector AopV in *A. citrulli* strain Aac5 encodes 342 amino acids with a predicted molecular size of 37 kDa. The homologous protein of AopV was found in *Acidovorax*, *Xanthomonas*, *Ralstonia*, and *Pseudomonas genomes*. The amino acid sequence of AopV shows 31% similarity to XopV from *Xanthomonas oryzae* pv. *oryzicola* (NCBI protein accession AEQ94858). AopV is most similar to XopV from *Xanthomonas*, compared with other homologous sequences from *Ralstonia* and *Pseudomonas*. The pairwise alignment of AopV and XopV is shown in Figure 1a. In addition, we found that representative AopV homologs from all four species contained nine consecutive identical amino acid sequences, which showed that these amino acid sequences are very conserved (Figure 1b).

### 2.2. ApoV Is a Type III Secreted Effector in A. citrulli

To determine if *A. citrulli aopV* was regulated by its T3SS, *aopV* gene expression levels were measured. The expression of *aopV* in the *hrpX* mutant [30] was significantly lower than that of the wild-type Aac5 (WT), which indicated the expression of *aopV* was regulated by the T3SS (Figure 2a). In addition, in order to test whether AopV was secreted during *A. citrulli* infection, secretion and translocation were tested with the adenylate cyclase (CyaA) translocation reporter assay. Eight hours after inoculation, the cAMP content in the host watermelon was significantly greater in the treatment inoculated with the strain WT-AopV-CyaA compared to the treatment inoculated with the strain Δ*hrcJ*-AopV-CyaA (Figure 2b). Together, these results showed that AopV is a T3E in *A. citrulli* strain Aac5.

### 2.3. AopV Is Expressed in the Nucleus and Cell Membrane of N. benthamiana

In order to determine the localization of effector protein AopV in plant cells, we transiently expressed AopV tagged with GFP (green fluorescent protein) in *N. benthamiana*, and observed it under laser confocal microscopy 48 h after transient expression. AopV showed green fluorescent signals in the cell nucleus and cell membrane (Figure 3). A cell nucleus marker carrying a red fluorescent protein (RFP) tag was used as the control for nucleus localization.

### 2.4. AopV Inhibits the ROS Burst and Expression Levels of PTI Marker Genes in N. Benthamiana

PTI is the plant’s first step in defense against pathogens, and an important PTI response is the ROS burst [1]. The transient expression of AopV significantly inhibited flg22-induced ROS compared to the empty vector (EV) control (Figure 4a). In addition, the effect of AopV on the expression of PTI marker genes was analyzed using qPCR. Compared with the EV control, AopV significantly inhibited the expression of PTI marker genes *NbPti5* and *NbAcre3* (Figure 4b), which suggested that the expression of AopV significantly inhibited the PTI pathway in *N. benthamiana*.

### 2.5. AopV Helps the Growth of D36E in N. benthamiana

*Pseudomonas syringae* pv. *tomato* (*Pst*) D36E colonized *N. benthamiana* expressing AopV at a significantly higher population level than *N. benthamiana* expressing EV (Figure 5). This indicated that AopV inhibited the PTI response induced by *Pst* D36E and enhanced *Pst* D36E’s colonization of *N. benthamiana*.

### 2.6. AopV Stimulates JA Production

JA is a main plant hormone involved in disease resistance. Some virulent pathogens have evolved various strategies to manipulate JA signaling to facilitate their exploitation of plant hosts [34,35]. To determine if AopV affects JA production, AopV and EV (control) were transiently expressed in *N. benthamiana*. After 48 h, the treated leaves were collected and JA production was measured. JA production in *N. benthamiana* was significantly increased in *N. benthamiana* expressing AopV (Figure 6).

### 2.7. AopV Interacted with ADT6

To further analyze the AopV interaction protein in the host, we screened an *A. citrulli*-induced yeast library of watermelon cDNA. The results showed an interaction between the aromatic dehydratase ADT6 from watermelon and the T3E AopV from *A. citrulli*. Aromatic dehydratase is involved in the biosynthesis of phenylalanine in *A. thaliana* [36]. The yeast two hybrid (Y2H) assay was conducted to verify this interaction. The Y2H assay showed that the test colony (pGBKT7-AopV and pGADT7-ADT6) and the positive control turned blue, but the negative control did not turn blue. We confirmed a direct interaction between AopV and ADT6 using this system (Figure 7a).

AopV-nluc and ADT6-cluc recombinant proteins were simultaneously expressed in *N. benthamiana*. Luciferase complementation imaging (LCI) analysis showed that the tissues expressing AopV and ADT6 produced fluorescence (Figure 7b). The test strains interacted to produce luciferase, which acted on the luciferin substrate to produce fluorescence. The negative control group did not produce fluorescence (Figure 7b). This further demonstrated the interaction between AopV and ADT6.

BiFC was conducted to provide more evidence for the interaction between AopV and ADT6. The pSPYNE^®^173-AopV and pSPYCE(M)-ADT6 recombinant proteins exhibited strong green fluorescence signals [37]. AopV and ADT6 were co-localized in the cell membrane of *N. benthamiana*, while the control group showed the red fluorescence marker, but no green fluorescence signal (Figure 7c). This indicated that there was a direct interaction between the effector protein AopV and ADT6.

## 3. Discussion

BFB is a devastating disease caused by *A. citrulli*, severely impacting the cucurbit industry worldwide. However, little is known about the molecular interaction between *A. citrulli* and host plants. *Acidovorax citrulli* secretes T3Es as a major pathogenicity strategy [28], however, only a few effectors have been studied in *A. citrulli* thus far. In 2020, Jimenez-Guerrero successfully identified some *A. citrulli* T3E proteins by machine learning and by comparing the transcriptomic differences between the *A. citrulli* wild-type strain M6 and the *hrpX* mutant strain [32]. In addition, two effectors AopP and AopN were identified and characterized in *A. citrulli* [20,33]. In this study, we found a putative T3E AopV in *A. citrulli* group II strain Aac5. The gene sequence from strain Aac5 is identical to that of *Aave_3085* in AAC00-1. BLAST analysis showed that AopV has homology with the *Xanthomonas oryzae* pv. *oryzicola* effector XopV, with 31% similarity (Figure 1a). The homologous protein of AopV was found in *Acidovorax, Xanthomonas, Ralstonia,* and *Pseudomonas* genomes. We found that nine amino acids in the homologous proteins of AopV in these strains are very conserved, and we speculate that these nine amino acids NDSGRFSEY from the AopV effector protein family may be a core motif performing important functions (Figure 1b). The importance and functions of these amino acids should be investigated in the future.

Screening key effector proteins is important. Our study showed that the *aopV* expression was down-regulated in *A. citrulli hrpX* mutant strain compared to the wild-type strain Aac5 (Figure 2a), indicating that the expression of *aopV* was regulated by T3SS. The regulatory gene *hrpX* is a key T3SS regulatory gene, and has been used for T3E screening in *A. citrulli* [30]. Additionally, the exocrine function is important for T3E identification. We used the CyaA translocation reporter assay to confirm that AopV in strain Aac5 had an exocrine function and was regulated by the T3SS (Figure 2b), which indicated that AopV was a T3E.

ROS plays a key role in both PTI and ETI responses. Transient ROS bursts can be part of early signaling, or ultimately leads to hypersensitivity, a type of programmed cell death that limits the systemic spread of pathogens [38]. We showed that AopV inhibited ROS production induced by flg22 in *N. benthamiana* (Figure 4a). Previous reports showed that T3Es AopP and AopN in *A. citrulli* also significantly inhibited ROS production [20,33]. ROS is an important signal molecule, and the burst of ROS is one of the early responses of PTI [39]. Another piece of evidence supporting AopV inhibiting PTI responses was provided by its ability to promote D36E colonization in *N. benthamiana* (Figure 5). D36E is a derivative strain of *Pst* DC3000 with all 36 T3Es deleted, but with intact T3SS and bacterial flagella that can stimulate PTI [40]. Strain D36E induced PTI responses in *N. benthamiana*, which were counter-inhibited by AopV. These results were consistent with previous findings of the effector XopV from *Xanthomonas oryzae* pv. *oryzicola*, which can inhibit the immune responses of plants [41]. In addition, AopV was predicted to have a nuclear localization signal at positions 206–234 (http://www.moseslab.csb.utoronto.ca/NLStradamus/, accessed on 25 September 2022), and our experiments demonstrated its localization in the nucleus. As opposed to AopV, although the homologous protein RipAD in *R. solanacearum* also has a nuclear localization signal, it is only detected in the cytoplasm and chloroplast [42]. 

Additionally, plant defense hormones are important immune responses in plants. ELISA and high-performance liquid chromatography (HPLC) can be used to detect phytohormones [20]. ELISA was used to detect JA in this study. Our study showed that after transiently expressing AopV in *N. benthamiana*, AopV significantly stimulated JA production (Figure 6). Our study is the first report on the effect of *A. citrulli* T3Es on JA. However, how AopV affects the JA pathway needs to be further explored. 

Gram-negative plant pathogenic bacteria deliver effector proteins into plant cells through the T3SS to manipulate the host cell environment [43]. However, in many cases, the targets of T3Es are unknown, including XopV from *X. oryzae* pv. *oryzicola* [44] and RipAD from *R. solanacearum* [42]. In our study, we found that AopV inhibited plant immune responses in *N. benthamiana*. To further study the interaction of *A. citrulli* with its natural host, watermelon, AopV was used as a bait to screen a yeast library of watermelon cDNA induced by *A. citrulli*, and a candidate target protein, ADT6, was identified. The interaction between AopV and ADT6 was confirmed by Y2H in vitro (Figure 7a) and bimolecular fluorescence complementation in vivo (Figure 7c). 

ADT6 is an aromatic dehydratase with 35% homology with ADT6 (NC_003070.9) in *A. thaliana*. In *A. thaliana,* ADT6 is involved in the synthesis of phenylalanine and participates in photosynthesis [36]. However, the role of ADT6 in watermelon is unknown. The interaction between AopV and ADT6 is different from the previously reported interaction between AopP and CLWRKY6 [20]. The interaction between AopV and ADT6 was in the cell membrane of *N. benthamiana* (Figure 7c), which indicated that ADT6 likely participated in different pathways than CLWRKY6, which occurred in the nucleus. The interaction between AopV and ADT6 was verified by LCI (Figure 7b). The experiment further confirmed that AopV from *A. citrulli* and ADT6 from watermelon directly interact. Through three verification methods, we determined that ADT6 from watermelon was an interaction target of *A. citrulli* T3E AopV. We speculated that ADT6 might be involved in the immune responses of watermelon. The specific immune pathways that ADT6 participates in are currently unknown and should be explored.

In summary, we found that AopV, a T3E protein of *A. citrulli*, was located in the cell membrane and nucleus of cells of *N. benthamiana*, and inhibited the PTI pathway by inhibiting the production of ROS in *N. benthamiana*. In addition, AopV affected the host immune responses by stimulating JA. Moreover, AopV interacted with ADT6 from watermelon. Our findings provide important insights into the immune responses of watermelon when interacting with *A. citrulli.*

## 4. Materials and Methods

### 4.1. Bacterial Strains, Plasmids, and Plant Materials

The strains and plasmids used in this study are listed in Appendix A. Watermelon (cv. ‘Ruixin’) and *N. benthamiana* seedlings were cultivated at 24 °C and 30~50% relative humidity. *Acidovorax citrulli* strains were cultured overnight at 28 °C in King’s B (KB) or T3SS induction medium [45] (Bacto peptone: 10 g, NaCl: 5 g, Yeast extract: 5 g, MgCl_2_·6H_2_O: 10 mmol/L, water: 1 L, pH = 5.8). *Escherichia coli* DH5α was cultured in Luria-Bertani (LB) medium at 37 °C for about 7 h, and *Agrobacterium tumefaciens* GV3101 was cultured in LB medium at 28 °C overnight. Antibiotics were used at the following concentrations: ampicillin 100 μg/mL, kanamycin 100 μg/mL, rifampicin 100 μg/mL, and chloramphenicol 50 μg/mL.

### 4.2. AopV Sequence Analysis

The amino acid sequences of AopV from *A. citrulli* and other homologous effector proteins were searched in NCBI, and the amino acids were aligned using ClustalW by DNAMAN version 5.2.2 (Lynnon Biosoft, Quebec, QC, Canada). 

### 4.3. CyaA Translocation Assay

The full-length cDNA of *aopV* (*Aave_3085* from *A. citrulli* AAC00-1, GenBank accession number CP000512.1) was cloned into the pBBRNolac-CyaA vector using seamless ligase (Vazyme), and then the constructed vector was transformed into the *A. citrulli* wild-type strain Aac5 and its T3SS-deficient *hrcJ* mutant by tri-parental conjugation [30]. The CyaA translocation reporter assay was used for effector identification analysis. All primers used are listed in Appendix A. *Acidovorax citrulli* strains were grown in KB medium to the logarithmic phase, resuspended in 10 mM MgCl_2_, and adjusted to 3 × 10^8^ CFU/mL. The leaves of 3-week-old watermelon seedlings were inoculated by syringe-infiltration. Two leaf discs (9 mm in diameter) from three leaves of each plant were taken 8 h after inoculation, and three different seedlings were sampled. An ELISA cAMP immunoassay kit (Enzo Life Sciences, Farmingdale, NY, USA) was used to detect cAMP levels.

### 4.4. Agrobacterium Infiltration

*Agrobacterium tumefaciens* GV3101 was infiltrated into leaves of 3- to 5-week-old *N. benthamiana* seedlings using a 1 mL syringe. The *A. tumefaciens* strains were cultured overnight at 28 °C in LB amended with kanamycin and rifampicin. The overnight cultures were centrifuged, resuspended in infiltration buffer (10 mM MES, 0.5 mM As, and 10 mM MgCl_2_), and adjusted to a final concentration equivalent to an optical density (OD_600_) = 0.5. After incubating in the dark for 1 h, the bacterial suspension was directly infiltrated into the *N. benthamiana* leaves.

### 4.5. ROS Burst Measurement

The full-length cDNA sequence of *aopV* was cloned and fused with pBI121-eGFP vector [46]. All primers used for PCR are listed in Appendix A. *Agrobacterium tumefaciens* GV3101 was used to transiently express EV (pBI121-eGFP empty vector) and AopV at an optical density (OD_600_) of 0.5. A Flg22-induced ROS assay was conducted with *N. benthamiana* leaves 36 h after inoculation. Twelve leaf discs (4 mm in diameter) were collected from each inoculation area and floated on 100 μL sterilized distilled water in a 96-well plate. The water was then removed, and 100 μL of a solution containing 100 nM flg22, 20 μg/mL horseradish peroxidase, and 100 μM luminol was added to each well. The luminescence was recorded immediately for 1 h using Tecan Infinite F200 luminometer (Tecan, Männedorf, Switzerland) [47].

### 4.6. RNA Extraction

To determine whether *aopV* was regulated by *hrpX*, RNA extraction and gene expression analysis were performed. The *A. citrulli* wild-type strain Aac5 and the *hrpX* mutant strain were cultured to the logarithmic phase, and then the cells were collected by centrifugation, resuspended in the T3SS induction medium and incubated for 3 h with shaking. RNA was extracted and reverse transcribed into cDNA using HiScript II Q RT SuperMix for qPCR kit (TIANGEN). qPCR was conducted to detect the expression of *aopV* in different strains. *rpoB* was used as a reference gene. The experiment was conducted three times.

To analyze the effect of AopV on the expression of PTI marker genes, RNA extraction and gene expression analysis were performed. RNA was extracted from *N. benthamiana* leaves 24 h after *Agrobacterium* infiltration, and reverse transcribed into cDNA using a kit (Zymo Research, Irvine, CA, United States; cat. no. R2024). The expression levels of the PTI-related genes *NbPti5* and *NbAcre31* were measured, and the *N. benthamiana* gene *EF1a* was used as the internal reference gene. The primers used in this experiment are shown in Appendix A. SYBR Green Real-Time PCR Master Mix (TOYOBO) was used for real-time quantitative PCR (qPCR). The experiment was conducted three times.

### 4.7. JA Detection

The full-length cDNA sequence of *aopV* was cloned into the PBI121-3FLAG vector, and the vector was transformed into *A. tumefaciens* GV3101. The *A. tumefaciens* GV3101 strains were infiltrated into three-week-old *N. benthamiana* leaves. *A. tumefaciens* GV3101 transformed with EV (pBI121-3FLAG) was used as a control. Four leaves were placed in a 15 mL centrifuge tube, and flash-frozen with liquid nitrogen 48 h after infiltration. JA content was measured using the ELISA method [48]. The experiment was conducted three times.

### 4.8. Subcellular Localization

The full-length cDNA sequence of *aopV* was cloned into the pBI121-eGFP vector. The *A. tumefaciens* GV3101 carrying AopV was expressed in three-week-old *N. benthamiana* seedlings*. A. tumefaciens* GV3101 carrying the EV was used as a control. RFP (red fluorescent protein) was used as a nuclear localization marker. The subcellular localization of the effector protein AopV was observed using a fluorescence confocal microscope (Zeiss LSM 880, Oberkochen, Germany) [49].

### 4.9. Impact of AopV on Pseudomonas Syringae pv. Tomato D36E Growth

The recombinant vector pBI121-AopV-3FLAG and EV were transiently expressed in *N. benthamiana* at a concentration of OD_600_ = 0.5. After four days, a bacterial suspension of *Pseudomonas syringae* pv. *tomato* (Pst) strain D36E (OD_600_ = 0.1) was infiltrated into *N. benthamiana* leaves. D36E is a derivative strain of Pst DC3000, in which all 36 T3Es were deleted, but the T3SS and the bacterial flagella that stimulate PTI were retained [40]. Six leaf discs were sampled (12 mm in diameter) 24 h after inoculation and plated on KB medium to measure D36E population levels in *N. benthamiana* leaves.

### 4.10. Yeast Two-Hybrid Assay

To analyze AopV and ADT6 interactions, a yeast two-hybrid assay was conducted. The full-length cDNA of AopV was cloned into the pGBKT7 vector, and the full-length cDNA of ADT6 (a putative ADT family protein encoded by *CM018018.1* from the watermelon 97103 genome; The ADT6 coding sequence is gene ID Cla97C01G011870 according to 97103) was cloned into the pGADT7 vector. Subsequently, AopV and ADT6 were co-transformed into the Y2H Gold strain. A positive control (Y2H Gold strain co-transformed with pGBKT7-53 and pGADT7-T) and negative control (Y2H Gold strain co-transformed with pGBKT7-lam and pGADT7-T) were used. The Y2H Gold strain was transformed with the above vectors and cultured on SD/-Leu-Trp medium [50]. Then, the transformants were cultured in YPDA medium (Peptone: 0.2 g, Yeast extract: 10 g, Glucose: 2 g, 2% Adenine solution: 15 mL) with shaking for 48 h, they were transferred to SD/-Leu-Trp-His-Ade agar medium containing 20 mg/mL X-α-gal. The blue colonies were considered to be potential positive interactions.

### 4.11. Luciferase (LUC) Complementary Imaging (LCI) Assay

LCI analysis was conducted as previously described with slight modifications. ADT6 was inserted into the pCAMBIA-cLUC vector and transformed into *A. tumefaciens* GV3101 [51,52]. AopV was inserted into the pCAMBIA-nLUC vector and transformed into *A. tumefaciens* GV3101. The two constructed *A. tumefaciens* GV3101 strains were equally mixed at a concentration of OD_600_ = 1.0, and co-expressed in *N. benthamiana*. After 48 h, leaves were collected and observed using a charge-coupled device (CCD) imaging apparatus (NightSHADE LB985; Berthold). The experiment was conducted three times independently.

### 4.12. BiFC Assays

The full-length cDNA sequence of *aopV* was cloned and fused with the pSPYNE^®^173 vector, and the full-length cDNA sequence of cloned ADT6 was fused with the pSPYCE(M) vector [37]. The proteins tested for interaction were constructed into *A. tumefaciens* GV3101 strains. They (AopV-nYFP and ADT6-cYFP) were equally mixed at a concentration of OD_600_ = 1.0, and co-expressed in *N. benthamiana*. The *A. tumefaciens* strains carrying either single empty or double empty vectors were used as controls. After 48 h, the fluorescence signal was observed under the excitation wavelength of 488 nm using laser confocal fluorescence microscopy (Zeiss LSM 880, Oberkochen, Germany). The experiment was conducted three times independently.

### 4.13. Statistical Analysis

Data were analyzed by one-way analysis of variance and Tukey’s honest significant difference tests. For the data analysis, the internal reference gene expression was set to 1, and the 2^−ΔΔCt^ method [53] was used. The independent-samples t-tests were used to analyze the significance of the differences between treatments. A *p*-value less than 0.05 was considered statistically significant. The statistical analysis was performed using GraphPad PRISM 5.0 software (GraphPad Software Inc., La Jolla, CA, USA).

## 5. Conclusions

By studying the function, expression, and secretion of AopV, we found AopV was a type III secreted effector protein that inhibited PTI responses. AopV was localized in the nucleus and cell membrane of *N. benthamiana*, and it inhibited ROS bursts in plants. More importantly, the effector protein AopV interacted with ADT6 from watermelon. The findings provide novel insights into how *A. citrulli* type III effector AopV interacts with its natural host watermelon, as well as with *N. benthamian.* How AopV interacts with ADT6 at the molecular level and its structure−function relationship are questions to be further investigated in the future.

## Figures and Tables

**Figure 1 ijms-23-11719-f001:**
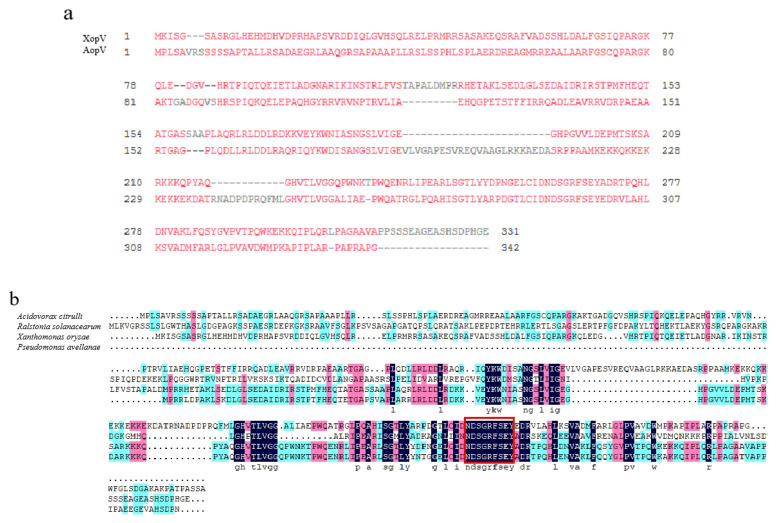
Amino acid sequence homology analysis. (**a**) Alignment of *Acidovorax citrulli* AopV and *Xanthomonas oryzae* pv. *oryzicola* XopV (NCBI protein accession AEQ94858). *Acidovorax citrulli* AopV shares 31% similarity with the sequence of the effector protein XopV of *Xanthomonas oryzae* pv. *oryzicola*. Sequences in red represent similar amino acids and gray denotes gaps. (**b**) Alignment of *A. citrulli* AopV and orthologs from *Xanthomonas oryzae* pv. *oryzicola* (AEQ94858), *Ralstonia solanacearum* (WP_003263919.1), and *Pseudomonas avellanae* (OZI86990.1). Amino acids highlighted in dark blue, cyan, and pink denote amino acids identical in four, three, and two species, respectively. Conserved amino acids among all four species are shown at the bottom in lowercase. The red box marks the nine consecutive identical amino acids.

**Figure 2 ijms-23-11719-f002:**
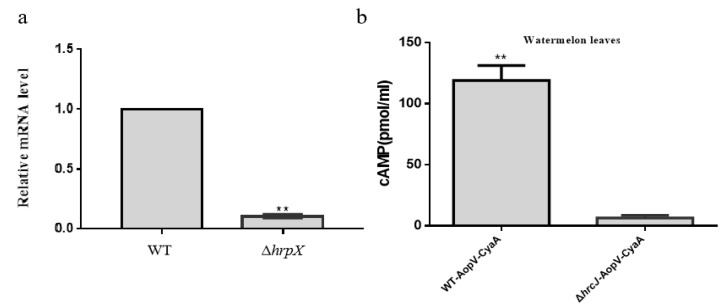
Identification of AopV as a type III secreted effector. (**a**) Quantitative real-time PCR was performed to determine whether *aopV* was regulated by *hrpX*. *RpoB* was used as an internal reference gene [30]. The experiment was conducted three times with similar results. The data represent the mean ± SD (standard deviation, *n* = 3). The ** above the bar indicates a significant difference determined by the *t*-test, *p* < 0.01; (**b**) Detection of cAMP content in watermelon host. *Acidovorax citrulli* wild-type strain Aac5 and Δ*hrcJ* carrying AopV-CyaA were inoculated into watermelon leaves and the cAMP contents were measured 8 h after inoculation. The experiment was conducted three times with similar results. The data represent the mean ± SD (*n* = 3). The ** above the bar indicates a significant difference determined by the *t*-test, *p <* 0.01.

**Figure 3 ijms-23-11719-f003:**
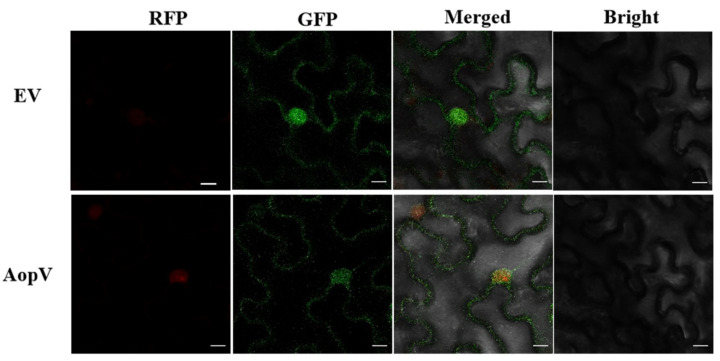
Subcellular localization of AopV in *Nicotiana benthamiana*. Leaves were injected with *Agrobacterium tumefaciens* GV3101 strain containing 35S::AopV-GFP (AopV) and observed at room temperature 48 h after infiltration using a confocal microscope (20×). Leaves injected with *A. tumefaciens* GV3101 containing 35S::EV-GFP were used as a control (EV). Red fluorescent protein (RFP) was used as a nuclear localization marker. The experiment was conducted three times with similar results. Scale bar, 20 µm.

**Figure 4 ijms-23-11719-f004:**
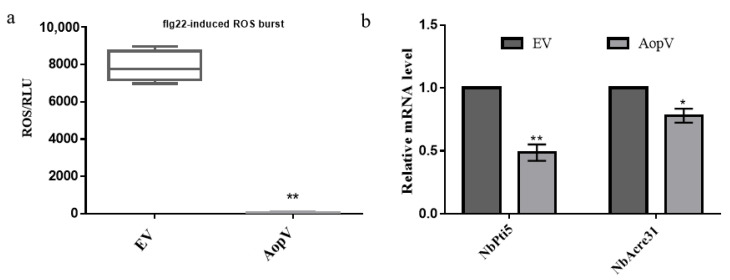
AopV inhibited PTI responses. (**a**) AopV inhibited flg22-induced ROS burst in *Nicotiana benthamiana*. Compared with *Agrobacterium tumefaciens* carrying an empty vector (EV), AopV-eGFP reduced ROS production in *N. benthamiana* induced by flg22. The experiment was carried out three times, and similar results were obtained each time. The data represent the mean ± SD (*n* = 10). (**b**) Expression of PTI related genes. Compared with *A. tumefaciens* carrying EV, AopV significantly inhibited the expression of PTI related genes *NbPti5* and *NbAcre3*. *EF1α* was used as an internal reference gene. The experiment was carried out three times each with similar results. The data represent the mean ± SD (*n* = 3). The * above bars indicate significant differences as determined by the *t*-test, *p* < 0.05. The ** means *p* < 0.01.

**Figure 5 ijms-23-11719-f005:**
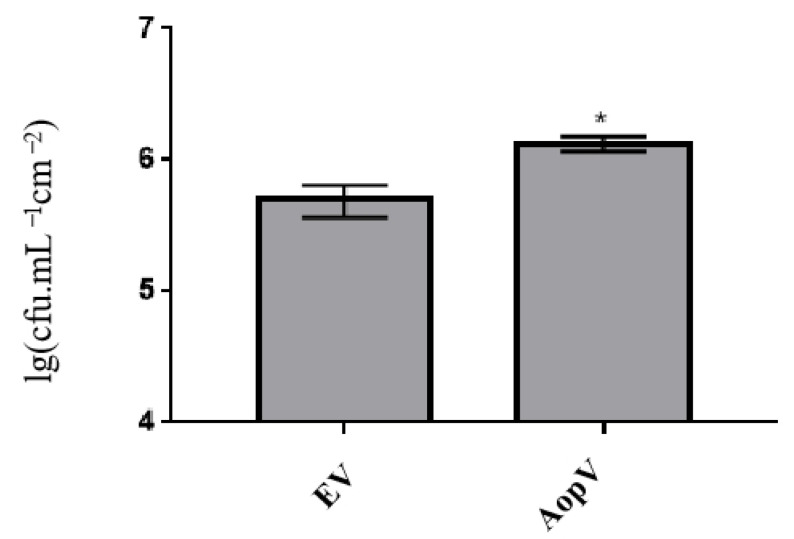
AopV helps the growth of *Pseudomonas syringae* pv. *tomato* (*Pst*) strain D36E in *Nicotiana benthamiana*. AopV was transiently expressed in *N. benthamiana*, and 4 days later, the bacterial suspension of *Pst* strain D36E was syringe-infiltrated. The empty vector (EV) was used as a control. After 24 h, the number of D36E colonies was counted. The experiment was carried out three times, and similar results were obtained each time. The data represent the mean ± SD (*n* = 3). * indicates statistically significant differences, as determined by the Student’s *t*-test, *p* < 0.05.

**Figure 6 ijms-23-11719-f006:**
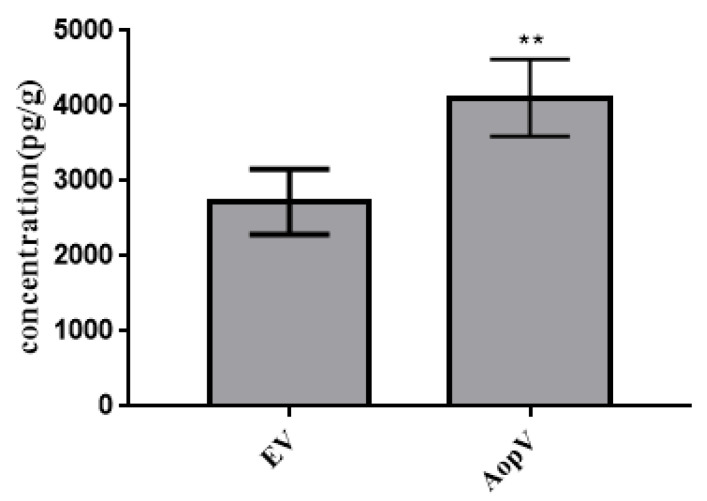
The effects of AopV on JA production in *Nicotiana benthamiana*. ELISA detection of JA content in plants. *Nicotiana benthamiana* leaves were injected with the *Agrobacterium tumefaciens* strain expressing AopV or empty vector (EV) at OD_600_ = 0.5. After 48 h, the leaves were collected and JA was quantified by ELISA. The experiment was carried out three times, and similar results were obtained each time. The data represent the mean ± SD (*n* = 3). ** indicates statistically significant differences, as determined by Student’s *t*-test, *p* < 0.01.

**Figure 7 ijms-23-11719-f007:**
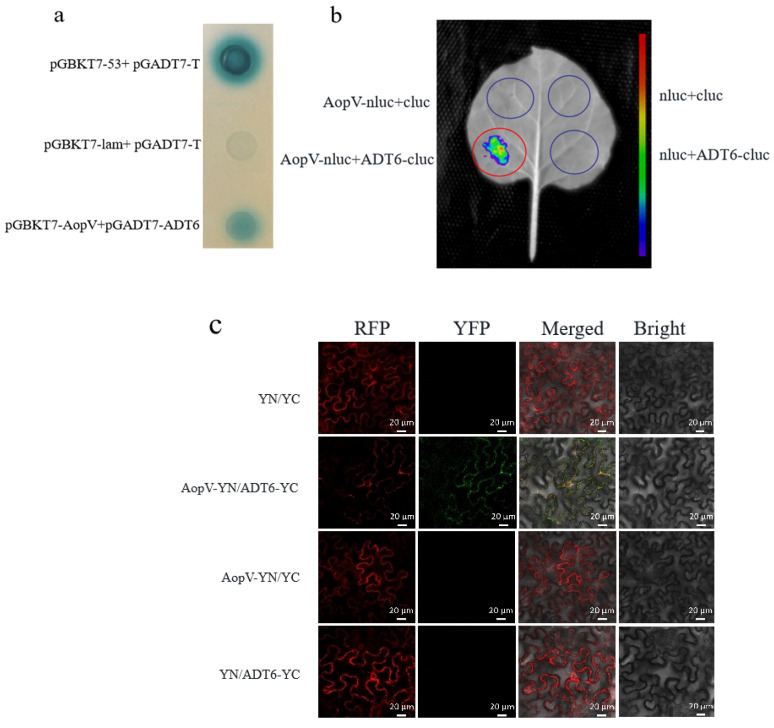
Interaction of *Acidovorax citrulli* effector AopV and watermelon aromatic dehydratase ADT6. (**a**) Yeast two-hybrid assay showed that AopV and ADT6 turned blue on SD/-Leu-Trp-His-Ade agar medium with 20 mg/mL X-α-gal. pGBKT7-53 and pGADT7-T are positive controls. pGBKT7-lam and pGADT7-T are negative controls. (**b**) Luc system verified protein interaction. *Agrobacterium tumefaciens* GV3101 carrying the AopV and ADT6 recombinant vector were expressed in *Nicotiana benthamiana*. After 48 h, the leaves were observed using a CCD imaging apparatus (NightSHADE LB985; Berthold). The experimental group produced fluorescence, and the control group showed no fluorescence. (**c**) Bimolecular Fluorescence Complementation (BiFC) verified the interaction between AopV and ADT6. AopV was fused with the nYFP tag, and ADT6 was fused with the cYFP tag. The subcellular localization of AopV and ADT6 in *N. benthamiana* leaves at 48 h after infiltration was observed under confocal microscopy. YN denotes pSPYNE^®^173 vector and YC denotes pSPYCE(M) vector. Red fluorescent protein was used as a cell membrane localization marker. Bars represent 20 µm. Each experiment was conducted three times, and similar results were obtained each time.

## Data Availability

Not applicable.

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
