# Peer review of "Acidovorax citrulli Effector AopV Suppresses Plant Immunity and Interacts with Aromatic Dehydratase ADT6 in Watermelon"

_ijms, 2022, doi:10.3390/ijms231911719_

Round 1
Reviewer 1 Report
This report (ijms-1928028) entitled “Acidovorax citrulli effector AopV suppresses plant immunity and interacts with aromatic dehydratase ADT6 in watermelon” by research group Yuwen Yang and Tingchang Zhao. This article is exciting and can make an excellent contribution to the scientific literature. However, the authors must address formal flaws and strengthen the presentation and clarity. The reviewer suggests a minor revision is needed before publication in a peer-reviewed journal. Some specific comments on the manuscript:
1. INTRODUCTION:
The authors suggest understanding the functions of A. citrulli T3Es will provide a basis for developing new strategies for BFB management. Need to explore new strategies for BFB management.
2. ABSTRACT:
The suggestion to the authors is to state the problem and share important results and suggest alternative methods or materials address the issue.
3. METHODOLOGY:
*The authors need to develop materials for controlling the disease of cucurbit plants caused by Acidovorax citrulli.
4. RESULTS AND DISCUSSION:
*Improve the scientific clarity, and mechanisms of virulence, and replace virulence term, it is used for viruses.
*Developing methods and materials and illustrating their mechanism of action is missing.
*Important suggestion is to improve flow, English. The overall presentation is descriptive and challenging to follow.
5. CONCLUSIONS:
*Do not duplicate the text of the abstract or results.
Author Response
Dear reviewer,
We sincerely appreciate your professional comments on our article. Your sincere suggestions and corrections are very valuable for the revision and improvement of our paper. According to your suggestion, we have carefully revised the original manuscript after careful consideration. Below are our responses to your suggestions and the corrections made in the article. All page and line numbers refer to the revised manuscript file.
Point 1: This article is exciting and can make an excellent contribution to the scientific literature. However, the authors must address formal flaws and strengthen the presentation and clarity. The reviewer suggests a minor revision is needed before publication in a peer-reviewed journal. Some specific comments on the manuscript:
- INTRODUCTION:
The authors suggest understanding the functions of A. citrulli T3Es will provide a basis for developing new strategies for BFB management. Need to explore new strategies for BFB management.
Response 1: Thank you for your affirmation of our work and your sincere suggestions. As for your comment that there is not sufficient evidence in this article to explore new strategies for BFB management according to understanding the functions of A. citrulli T3Es, we agree with your comment. Our original intention is to put forward a possibility based on the current experimental results. We have modified the expression of this part. We also indicate that we only proposed a possibility based on the existing experimental results and we will continue to carry out further research on it in the future. In addition, some references have been added to the research progress section of A. citrulli (line 77-87).
- ABSTRACT:
The suggestion to the authors is to state the problem and share important results and suggest alternative methods or materials address the issue.
Response 2: Thank you for your sincere suggestions. We have made significant changes to the summary section. We have shared the important results of the experiments and revised the abstract section.
- METHODOLOGY:
*The authors need to develop materials for controlling the disease of cucurbit plants caused by Acidovorax citrulli.
Response 3: Thank you for your sincere suggestions. We revised the statement that “explore new strategies for BFB management according to understanding the functions of A. citrulli T3Es”. We study the function of Acidovorax citrulli effector proteins and their targets, and explore their interaction mechanism. Our research is still far from developing materials to prevent BFB, but we hope understanding molecular interactions between pathogen and host will provide the fundamental basis for future disease management.
- RESULTS AND DISCUSSION:
*Improve the scientific clarity, and mechanisms of , and replace virulence term, it is used for viruses.
Response 4: Thank you for your sincere suggestion. The expression “virulence” appeared three times in total. One was in the abstract, we changed the original sentence “Although BFB has great destructive potential and economical importance, the molecular mechanisms of virulence of A. citrulli are not clear” to “Although BFB has great destructive potential and economical importance, the molecular mechanisms of pathogenicity of A. citrulli are not clear” (Page 1, Line 16-17). The second was in the introduction (Page 2, Line 92). The third was in the discussion. We replaced “virulence” with “pathogenicity”.
*Developing methods and materials and illustrating their mechanism of action is missing.
Response: Thank you very much for your correction. We have revised and supplemented this part. In the discussion part, the similarities and differences with previous works are compared.
*Important suggestion is to improve flow, English. The overall presentation is descriptive and challenging to follow.
Response: Thank you for your sincere suggestion. We have added more discussion of these concerns in the Discussion. We regret there were problems with the language. The paper has been carefully edited and revised by a native English speaker to improve the flow, grammar, and readability.
- CONCLUSIONS:
*Do not duplicate the text of the abstract or results.
Thank you for your sincere suggestions. In the conclusion of this paper, we have added our prospective work at the end of the conclusion and modified the sentence according to your suggestions.
We would like to express our sincere thanks again to you for spending time and energy to review our manuscript selflessly, and making such valuable comments on our work.
Reviewer 2 Report
The paper “Acidovorax citrulli effector AopV suppresses plant immunity and interacts with aromatic dehydratase ADT6 in watermelon” focused on characterizing a novel type three effector (T3E) AopV in A. citriulli using strain Aac5 as well as understanding its mode of interaction with the host. Authors confirmed the existence of homologues of this gene in other A. citrulli strains. T3Es mediate bacterial interaction with plant, promoting bacteria colonization and disease progress. Where they mediate gene for gene interaction, characterization of such effectors and host genes are important in identification of novel hypersensitivity responses and resistance genes. Effector AopV has not been characterized previously and this paper is the first to describe it. I am especially fascinated by the depth of standard experiments and presentation.
Confirmation of T3S was through qPCR by using hrpX mutant and checking for the expression of aopV. While a gene reporter assay coupling the N-terminus to another protein would have provided more support, this experiment and other evidences confirm that it is a type three secreted effector. Other experiments carried out to demonstrate localization as well as host interaction improved the understanding of this novel effector. Ordinarily, a yeast two hybrid assay should confirm the interactions of AopV with ADT6. The authors further improved this result by Luciferase complementation imaging as well as bimolecular fluorescence complementation, both of which confirmed Y2H assay.
The paper is well-written and the conclusions drew from many standard experiments , as demonstrated in many previous studies including some from this lab such as https://www.frontiersin.org/articles/10.3389/fpls.2020.579218/full and https://www.frontiersin.org/articles/10.3389/fsufs.2022.995894/full.
Minor corrections
Line 99: change “was shown” to “is shown”
Line 142: write EV out in full before abbreviating
The labels of Figures 7a and 7b could be improved.
Author Response
Dear reviewer,
We would like to express our sincere thanks to you for spending time and energy to review our manuscript selflessly in your busy schedule, and providing valuable comments and suggestions. We revised the manuscript according to your comments. The detailed correction is listed below. The page and line numbers refer to the revised manuscript file.
Point 1:Confirmation of T3S was through qPCR by using hrpX mutant and checking for the expression of aopV. While a gene reporter assay coupling the N-terminus to another protein would have provided more support, this experiment and other evidences confirm that it is a type three secreted effector.
Response 1:Thank you for your sincere suggestions. We did the adenylate cyclase (Cya) translocation reporter assay. The Cya assay results were described in page 3 and the Cya assay method was described in page 9 of the article.
Point 2:Minor corrections
Line 99: change “was shown” to “is shown”
Line 142: write EV out in full before abbreviating
The labels of Figures 7a and 7b could be improved.
Response 2: Thank you very much for your affirmation of our work and your correction. We have corrected the above problems and corrected the labels of Figures 7a and 7b. We have carefully checked other parts of the full text that use these expressions to ensure that there is no such error. The specific revisions are as follows:
Line 99 (Page 2): Revised “was shown” to “is shown”.
Line 142 (Page 2): Revised “EV” to “empty vector”.
Line 172 (Page 5): Revised “EV” to “empty vector”.
We would like to express our sincere thanks again to you for spending time and energy to review our manuscript selflessly, and making such valuable comments on our work.